# Effects of Social Isolation Measures Caused by the COVID-19 Pandemic on Occupational Balance, Participation, and Activities’ Satisfaction in the Spanish Population

**DOI:** 10.3390/ijerph19116497

**Published:** 2022-05-26

**Authors:** Cristina Rodríguez-Rivas, Lucia Rocío Camacho-Montaño, Cristina García-Bravo, María García-de-Miguel, Marta Pérez-de-Heredia-Torres, Elisabet Huertas-Hoyas

**Affiliations:** 1Hospital Fundación Instituto San José, 28054 Madrid, Spain; c.rodriguezr.to@gmail.com; 2Department of Physical Therapy, Occupational Therapy, Rehabilitation and Physical Medicine, Rey Juan Carlos University, 28922 Madrid, Spain; lucia.camacho@urjc.es (L.R.C.-M.); marta.perezdeheredia@urjc.es (M.P.-d.-H.-T.); elisabet.huertas@urjc.es (E.H.-H.); 3Centro Integral de Atención Neurorrehabilitadora, Grupo 5, 50017 Zaragoza, Spain; mgm_96@hotmail.es

**Keywords:** activities of daily living, COVID-19, health, occupational balance, pandemic

## Abstract

The COVID-19 pandemic caused a lot of social and health chaos. Our main aim in this study was to examine the impact of the COVID-19 pandemic on occupational balance in the Spanish population, one year post the beginning of the pandemic compared with the pre-pandemic period. Data were collected among the Spanish population over 18 years of age by the Occupational Balance Questionnaire online survey; questions about satisfaction and performance of activities, and on the modification of routine and habits were asked. A total of 300 participants were included; 55.3% were female, with a mean age of 41.39 years. Significantly greater occupational imbalance was found in 1-year-post-confinement period of pandemic, as well as an increase in the difficulty of performance and a decrease in satisfaction with it. A greater number of the sample had modified their routines (*p* < 0.01), lost habits (*p* < 0.01), and did not resume habits (*p* < 0.01). In the analysis by age groups, differences were found in the variables related to habits and occupational balance. The social restriction measures negatively impacted occupational balance in the Spanish population. There was a decrease in occupational participation, increased difficulty in performance, decreased satisfaction in occupational performance, and modification of habits and routines.

## 1. Introduction

The World Health Organization (WHO) considered the coronavirus disease 2019 (COVID-19) a global pandemic in March 2020 [1,2]. It is known that symptoms of COVID-19 lead to health risks, such as social isolation, depression, and generalized insecurity [3,4,5,6]; these alterations may significantly affect daily activities, social interactions, and sense of well-being [7]. In Spain, several public health social measures were implemented, such as home confinement, urban mobility limits, restaurants restrictions, nightlife hours restrictions, teleworking, the requirement of COVID-19 passports to access bars, restaurants, and hospitals, leisure and cultural activities restrictions, and so on. Now, two years later, citizens are still requested to practice some restrictive measures such as mask wearing and social distancing to prevent cross contamination.

Home confinement, forced by the pandemic has indirectly had a major impact on daily activities, habits, and routines [8,9,10], which negatively impacts life satisfaction [11]. In this sense, several authors [11,12,13] have described significant changes in routines, with reduced or lost leisure and recreational activities, reduced physical activity, restricted participation, and psychological disturbances. Previous studies [14] have related lifestyles, routines, and daily activities to health, whereby an alteration in these patterns can lead to the development of risk factors of various diseases. Consistent with this, the abrupt interruption of routines and social isolation can disrupt the balance between all daily activities, including basic and instrumental, work, education, leisure, and social participation, affecting their well-being [14,15]. Occupational balance (OB) is defined as the individual’s own perception of the appropriate amount and variety of activities in which they participate, as well as the time devoted to each of the activities [16]. Thus, participation in activities in a balanced and satisfying way contributes to the maintenance of health and well-being [14,17]. Previous research [14,15,18] support a relationship between OB, health, and life satisfaction. Therefore, maintaining a balance in daily routine is an essential need, even in difficult situations [15].

Some authors, including González-Bernal et al. [15] found a correlation between OB and physical and mental health during home confinement. However, no research has been conducted on the impact of social restriction measures caused by COVID-19 on people’s health and OB for one year, over the longer terms, or over longer periods of confinement. Thus, the main purpose of this study was to examine the impact of social isolation measures caused by the COVID-19 pandemic on OB in the Spanish population one year after the beginning of the pandemic compared with that of pre-pandemic periods.

Additionally, this study will also determine the existence of differences in OB between sex, age, and region of residence.

## 2. Materials and Methods

### 2.1. Design

A cross-sectional, descriptive design was used, following the guidelines of the Strengthening the Reporting of Observational Studies in Epidemiology (STROBE) checklist [19]. The initiative STROBE developed recommendations on what should be included in an accurate and complete report of an observational study [19].

This study was approved by the Rey Juan Carlos University Research Ethics Committee, reference number 3011202021520. All participants were informed in advance regarding the procedures, risks, and benefits of the research. In addition, informed consent forms were requested to be signed and provided at the start of the study, in accordance with the Declaration of Helsinki, as revised at the World Medical Association Assembly in 2013 [20]. The informed consent was created in the beginning of the survey, which was designed using the Microsoft Forms program. It was sent by WhatsApp and Facebook to the participants.

### 2.2. Participants

A convenience and snowball non-probabilistic sampling approach as used to recruit participants. Participants were recruited over a period of one month (19 February–20 March 2021) through social media (Facebook and WhatsApp).

The inclusion criteria for participants were (1) people aged 18 years of age or older; and (2) residents of Spain. The exclusion criteria were (1) inability to respond to the questions asked; and (2) the presence of other diseases arising after February 2020, that caused an alteration to occupational balance, performance of daily activities, routines, and habits.

### 2.3. Data Collection

Data were collected through an online survey created with the Microsoft Forms application from Office 365 (Microsoft Office 365). Socio-demographic data were collected (sex, age, and region of residence), as well as data on activity balance.

#### 2.3.1. Outcome Measures

##### Occupational Balance Questionnaire (OBQ)

The OBQ is a reliable instrument in which the person assesses their OB in relation to their situation in their daily life (in the pre-pandemic period and during the pandemic) [21]. It consists of 13 statements exploring OB, meaning, time use, and satisfaction. It is scored on a six-digit scale, with values ranging from 0, “strongly disagree”, to 5, “strongly agree”. A total score between 0 and 65 can be obtained, in which the higher the score, the higher the OB. It was adapted and validated for the Spanish population (OBQ-E), showing good psychometric properties [22].

##### Questions about Satisfaction and Performance of an Activities

For the design of these questions, the Canadian Occupational Performance Measure instrument was considered [23]. The activities were distributed by Occupational Performance Areas: Self-care, Productivity, and Leisure. For of Self-care, examples of basic activities are distinguished, including those pertaining to personal self-care and functional mobility, and instrumental activities of daily living. As for Productivity, we included paid or unpaid work, household chores, and tasks in the study. Finally, for Leisure, examples include passive, active, and social recreational activities. To learn the activities in which the participant presents difficulty in performance, he/she is asked to select one or more of the above-mentioned activities. In the case of not having presented difficulty, the participant selected the option “I do/did not possess any difficulty”. In order to obtain information about the degree of satisfaction in the performance of activities, a scale composed of 5 values was used: 1, “no satisfaction at all”; 2, “little satisfaction”; 3, “medium satisfaction”; 4, “great satisfaction”; and 5, “much more satisfaction”. Participants select one value for each activity. The higher score, the greater degree of satisfaction in occupational performance. 

##### Questions on the Modification of Routine and Habits

These questions were designed by the authors in the Linkert style. Regarding routine, the following was proposed: “Do you consider that, as a result of COVID-19, your routine has changed?". There is only the possibility of a “Yes” or “No” answer. In relation to habits, questions were asked about the loss of or reduction in some habits and the retaking and acquisition of new ones, with a “Yes” or “No” answer. In addition, there was the possibility of providing examples of each one of them. These questions were formulated with the objective of learning the changes caused due to the pandemic in habits and routine.

The questions were oriented towards two points in time, the period prior to the declaration of a global pandemic (March 2020) and one year after the pandemic, confinement, social isolation measures declaration (February to March 2021).

### 2.4. Data Analysis

The estimated effect size for the main outcome measures established in the present study was 0.25. Considering a power of the statistical test of 0.80 and an alpha error of 0.05 for the pre-post comparison for independent samples, a minimum of 159 subjects (+15% for losses: 182) was required for the present study, according to the G*Power Software (version 3.1.7).

Descriptive data were presented by means of the frequency of the categorical variables and the mean and standard deviation of the continuous variables. Non-normal distribution was confirmed by the Kolmogorov–Smirnov test. Differences between the pre-pandemic period and during pandemic were analyzed using the Wilcoxon test for related samples on quantitative variables. A non-parametric method was used to study the differences between independent samples (sex/age range/regions of Spain) using the Mann–Whitney U or Kruskal–Wallis test. The *p* value is *p* < 0.05.

The analysis of the variables was performed with the statistical program IBM SPSS Statistics for Windows, V.27.0 (Copyright 2013 IBM SPSS Corp, Armonk, NY, USA).

## 3. Results

The final sample was composed of 300 participants, 55.3% female, with a mean age of 41.39 ± 18.743 years. The descriptive data of the sample can be found in Table 1. Overall, the percentage of missing data was <5%, and no special handling of the missing data was conducted.

When analyzing the differences between the OBQ-E variable before the pandemic (45 ± 9.52) and post-confinement period of the pandemic (34.89 ± 12.81) in the complete sample (through the Wilcoxon test), significant differences were found in all variables with values of greater impact in those referring to the post-confinement period of pandemic. In other words, greater occupational imbalance (Z = −10.789; *p* < 0.001) was found.

Regarding the analysis according to the groups as independent samples (Table 2), women had significantly higher OB before the pandemic; nevertheless, they decreased and ceased to be significant during the pandemic. If the analysis was performed according to age groups (young, 18–30 years of age; adult, 31–64 years of age; and older adult, 65 years of age and older), differences were found. The younger age group had more unfavorable scores. The best scores of OB were obtained by the older adult group. In terms of region of residence, people in the south had the lowest OB; before the pandemic these differences were significant, but not during the pandemic (Table 2).

Regarding habits and routines, significant differences were found between pre-pandemic and during the first year of pandemic in the whole sample, with a greater number of the sample having modified their routines (*p* < 0.01), lost habits (*p* < 0.01), and not resumed habits (*p* < 0.01) (Figure 1).

When analyzed by age group, young people (18–30 years of age), adults (31–64 years of age), and older adults (65 years of age and older), significant differences were found in the acquisition of new habits, with young people being the most likely to acquire new habits (X^2^ = 7.932; *p* < 0.019) (Figure 2).

Regarding performance and satisfaction, when analyzing the differences between the variables before the pandemic and one year post the beginning of pandemic, significant differences were found in the whole sample, with the most affected values being those referring to the post-confinement period of pandemic, i.e., greater difficulty in performance (Figure 3) and lower satisfaction (Figure 4a–c). Significant differences were found in satisfaction with daily activities performance (X^2^ = 45.87, *p* = 0.001), in satisfaction with productivity performance (X^2^ = 313.98, *p* = 0.001), and in satisfaction with leisure (X^2^ = 50.97, *p* = 0.01).

## 4. Discussion

The purpose of this study was to examine the impact of social isolation measures caused by the COVID-19 pandemic on OB in the Spanish population one year after the beginning of the pandemic compared with the pre-pandemic period. Additionally, this study was conducted to determine the existence of differences in OB between sex, age, and region of residence. Our results indicate that significantly greater occupational imbalance was found in 1-year-post-confinement period of pandemic; an increase in the difficulty of performance and a decrease in satisfaction with it. A greater number of the sample had modified their routines, lost habits, and did not resume habits. In the analysis by age groups, differences were found in the variables related to habits and occupational balance.

The results showed an increase in occupational imbalance during the social isolation measures period. Some of the activities were reduced, as well as the time spent on them. This is consistent with a review by Eklund et al. (2017) [24], in which they indicated that, to achieve or maintain OB, a balanced and varied mix of occupations is necessary, considering not only the number of activities, but also the time spent on each activity. These results were similar to that found in a study by Lipskaya-Velikovsky et al. (2021) [25], where they found that at least half of the participants had interrupted their routines, thus reducing activities, and only a limited range of activities had been maintained. The study by Gonzalez-Bernal et al. (2020) [15] found that age, the perception of having received enough information, not telecommuting, and not being infected by COVID-19 contributed to a better occupational balance, coinciding with our results, since our data indicate that older people are the ones who best maintained an occupational balance. Moreover, Savitsky et al. (2021) [26] found a positive correlation between occupational satisfaction, the importance of the activities in which they participated, workload, professional support, and psychological reward.

Before the pandemic, women had a significantly greater balance of activities than men, but these differences became similar during the pandemic and were no longer significant. Nevertheless, these data are at odds with pre-pandemic research, which reported no significant occupational balance differences between sexes [25].

In terms of OB, significant differences were observed based on age. The older adult group had the highest OB. Young people, both in the period before the pandemic and during the later period of the pandemic, had higher occupational imbalance than adults and older adults. In contrast to our results, Håkansson et al. [27] found that age had no influence on OB. However, the data are similar to those found in the study by González-Bernal et al. (2020) [15], who reported a positive correlation between OB and age. Nevertheless, this correlation was weak; therefore, given these circumstances and the lack of sufficient articles supporting the fact that these variables are related, we cannot generalize these results. This supports the initial hypothesis of the presence of occupational imbalance in the wake of the COVID-19 pandemic. The differences found with respect to occupational balance can be related to the loss of habits being greater in the young people and adult population, 87.5% and 91.9%, respectively. This can be related to the presence of resumed or acquired habits, facing the modification of routines, and reflecting an adaptive attitude to the new circumstances. Thus, in the young people, it is observed that 32.5% of participants resumed habits and 53.8% acquired new ones; in adults, 31.3% resumed and 43.4% acquired new ones; compared with 28.6% who resumed and 25.7% who acquired new ones in the older adult group. However, there are more participants who have lost or reduced their habits (87.5% in the young people, 91.9% in adults, and 85.7% in older adults) than those who have taken up or acquired new habits, thus supporting the initial hypothesis of the presence of occupational imbalance as a result of the COVID-19 pandemic. They are also the group that has acquired or resumed fewer habits. However, no studies have found a justification or reflection on the modification of routines because of the pandemic and related to age.

As with the sex variable, the differences between the two time periods were not significant, and subsequently equalized and were no longer significant. The north had the best OB, followed by the south before the pandemic, and the center during the pandemic. Therefore, the southern zone was the most affected during confinement. No studies have been found that compare the possible justifications for place of residence. Future studies can focus on these aspects, as our results indicate differences that may have a negative impact on the health and OB of both the general population and people affected by COVID-19.

Regarding participation and satisfaction of activities, a decrease in participation and satisfaction was observed in all three areas of occupational performance. According to the study conducted by Savitsky et al. (2021) [26], who analyzed job satisfaction of nurses during the pandemic, there is a positive correlation between occupational satisfaction, the significance of the occupation, workload, professional support, and psychological reward. Furthermore, it alludes to the importance of occupational satisfaction for psychological and physical well-being. As alluded to by Håkansson et al. (2021) [28] a strong association was found between OB and stress, being null or insignificant in those with balance. In turn, those with the highest OB scores were those who had social support at work, participated in leisure and recreational activities, and spent time resting. Considering our study, it is possible to think that the decrease in satisfaction is due to the impossibility of participating in meaningful activities and the decrease in the number of activities performed.

Most participants agreed that they experienced a change in routine, with some of their habits being lost or diminished, while less than half of the participants reported having taken up or acquiring a new habit. Such data are consistent with the study conducted by Lipskaya-Velikovsky et al. (2021) [25], which aimed to analyze the impact of COVID-19 on daily life and mental health in the healthy population of Israel during home confinement. They found that at least half of the participants disrupted their routines, reducing activities, and only a limited range of activities were maintained. However, these data do not coincide with the study carried out by Di Renzo et al. [29], which analyzed dietary and lifestyle changes in the Italian population during COVID-19 related confinement. Regarding lifestyle, most of the population reported no changes in their habits, with a minority indicating that their habits had worsened or improved.

### Limitations

This study has some limitations. Despite presenting a large sample there may not be enough of a population to establish a generalization of the results. Due to the circumstances in which we found ourselves during the study, characterized by mobility restrictions, we used an online survey, a usual methodology during the pandemic period [29,30,31]. This method may have given rise to bias, as it is self-reported and not an objective measure. In terms of sample selection bias, this sample reflects a high number of young people and adults compared with older adults, with the latter being the most difficult to access. Regarding the social desirability bias, we tried to mitigate it by making the survey anonymous. Moreover, the measurement of the variables, referring to two different points in time and carried out on the same occasion may have given rise to memory bias. However, this study also presents important strengths. It is the first time that such a situation has occurred worldwide, affecting people of diverse nature (age, sex, and place of residence) from which quantitative and qualitative information can be obtained remotely. This online interview methodology made it possible to analyze how social isolation affects people’s occupational balance. This contributes to improve the information and design of the intervention of immunodeficient pathologies which, due to their clinical conditions, also require processes of social isolation.

The lack of evidence on the subject made it difficult to compare results; therefore, future studies are needed in this area. Likewise, more longitudinal studies are needed to identify possible long-term consequences, as well as in samples of similar clinical conditions.

## 5. Conclusions

This study provides considerable evidence on the impact of social isolation measures caused by the COVID-19 pandemic on OB in the Spanish population. A deficit in OB was found. These disturbances are susceptible to intervention since, as previously described, acting on the imbalance between activities and routines can promote a better well-being. This study may contribute to improve the development of intervention protocols for people who require social isolation processes. For example, immunodeficient pathologies, which due to their clinical conditions also require it. The lack of evidence on the subject reflects the necessity of more longitudinal studies, as well as the need to identify possible long-term consequences.

## Figures and Tables

**Figure 1 ijerph-19-06497-f001:**
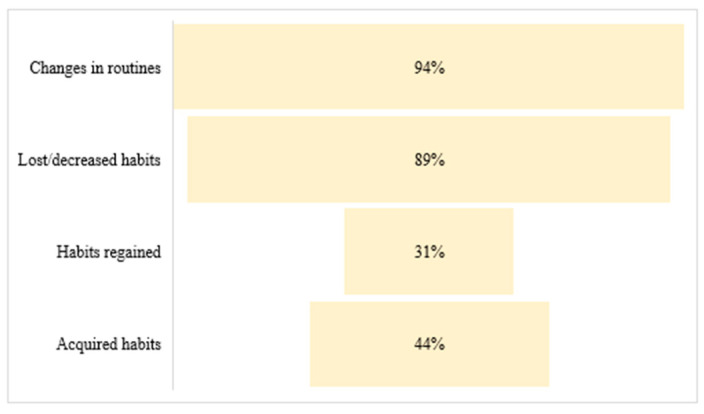
Changes in routines and habits during the global pandemic by COVID-19.

**Figure 2 ijerph-19-06497-f002:**
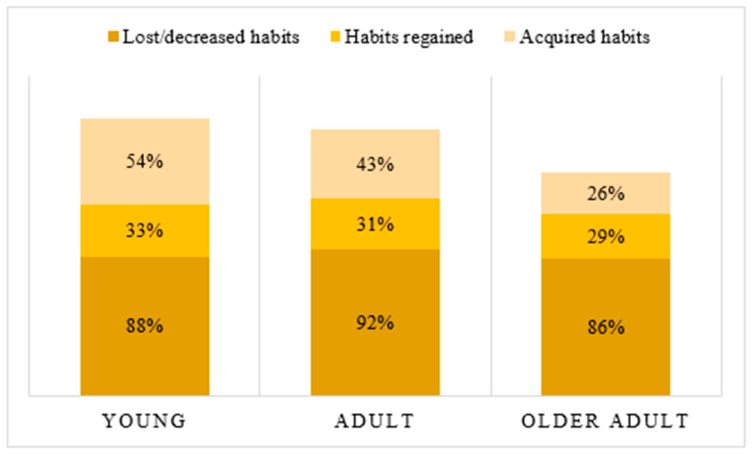
Analysis of age-related change in habits.

**Figure 3 ijerph-19-06497-f003:**
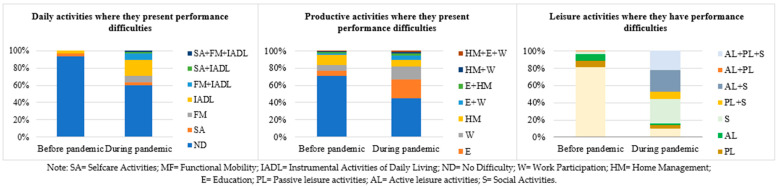
Activities where they present performance difficulties.

**Figure 4 ijerph-19-06497-f004:**
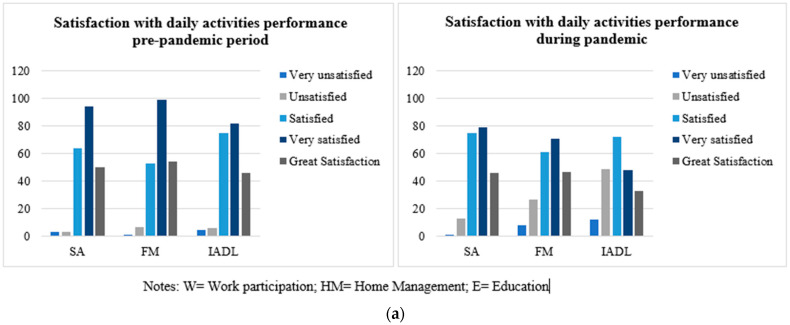
(**a**) Satisfaction with daily activities performance. (**b**) Satisfaction with productivity performance. (**c**) Satisfaction with leisure.

**Table 1 ijerph-19-06497-t001:** Socio-demographic characteristics of the sample (*n* = 300).

Variables	Descriptive Data
**Sex**	
Male	134 (44.7%)
Female	166 (55.3%)
**Age (years)**	
Median (IQR)	39 (56–23)
**Age range**	
18–30 years of age	143 (47.7%)
31–64 years of age	118 (39.3%)
65 years of age or more	37 (12.3%)
Missing data	2 (0.7%)
**Region of residence**	
North	65 (21.7%)
Center	213 (71%)
South	19 (6.3%)
Missing data	3 (1%)

**Table 2 ijerph-19-06497-t002:** Analysis of differences between independent samples in EuroQol 5D-5L and Occupational Balance Questionnaire.

	OBQ-E		
	Median (IQR)	Statistic Value	*p*
**Whole sample**		−10.789	0.001
before pandemic	45 (51–39)		
during pandemic	35 (43.75–26)		
**Sex**			
before pandemic			
Man	44 (50–37)	9272.000 ^a^	0.023
Woman	46 (52–40)		
during pandemic			
Man	35 (43–24.75)	8907.000 ^a^	0.33
Woman	36 (44.25–26.75)		
**Range of age**			
before pandemic			
18–30 years of age	43 (50–37)	28.396 ^b^	0
31–64 years of age	46 (52–40)		
65 years of age or more	50 (57–46)		
during pandemic			
18–30 years of age	32 (40–24)	20.511 ^b^	0
31–64 years of age	39 (47–26)		
65 years of age or more	42 (49.5–36)		
**Region of residence**			
before pandemic			
North	48 (53–43)	6.756 ^b^	0.034
Center	44 (51–38)		
South	45 (50–37)		
during pandemic			
North	38.5 (47.75–26)	2.047 ^b^	0.359
Center	35 (43–26)		
South	35 (39.5–24)		

Notes: M: mean; SD: standard deviation; OBQ-E: Occupational Balance Questionnaire; *p* values were calculated using the ^a^ Mann–Whitney U test or ^b^ Kruskal–Wallis test.

## Data Availability

The database can be accessed on request from the corresponding author.

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
