# Peer review of "Effects of Social Isolation Measures Caused by the COVID-19 Pandemic on Occupational Balance, Participation, and Activities’ Satisfaction in the Spanish Population"

_ijerph, 2022, doi:10.3390/ijerph19116497_

Round 1
Reviewer 1 Report
This is a very interesting article about a serious issue that is COVID-19 pandemic.
Overall, the article is well written and provides interesting insights on the impact of the pandemic on the satisfaction of Spanish population regarding occupational activities.
However, I feel like some aspects of the manuscript could be improved. Below, you can find my suggestions that I hope they can be helpful. Also, it would also be of your benefit to have the manuscript reviewed by an English-speaking individual.
Page 2, lines 73-74: As the data was collected online, could you explain in detail how informed consent was obtained (e.g.: a informed consent form in the beginning of the survey, informed consent sent by email to the participants…)
Page 2, line 85: Is that “Result” supposed to be there?
Page 2, lines 86-88: This sentence seems to be out of place.
Page 3, line 139: How many participants were you aiming to target? How many people answered the survey? Were there any participants who were excluded from the analysis?
Page 4, table 2: You should provide the categories in English (please, confirm in the other tables and figures that all the legends are in English as well)
Page 5, line 174-175: What was the rationale behind the age cut offs? In several studies, young adults are defined as individuals below 35 years of age. It seems a little odd to me that individuals between 31 and 64 fall under the category of young adults so I was wondering what was the rationale behind the classification. Also, what is the difference between the terms “young adults” and “young people”?
Page 5, figure 2: (Related to the previous comment) the classification presented in this figure seems more accurate, but due to the explanation provided before, the reads may be confused about who are “Middle aged” people described in the figure.
Page 6, Figure 4.1: This is just a suggestion, if possible, try to use shades of grey that are not so similar because the bars of Satisfied and very satisfied are very similar in terms of colour. (I am aware that this may be difficult and it may also be easy to understand due to the order of the bars being the same as the order of the notes on the side, hence this is just a suggestion)
Page 7, line 213: It would be interesting to provide a line or two that would summarize the main results of the study at the end of the first paragraph of the discussion.
Page 8, line 222: How can you explain the differences between the results of your study and those of Gonzalez-Bernal et al. (2020) study?
Page 8, line 229: “One similar, but contrary,” could you write this part of the sentence in a clearer way? It sounds a bit confusing for the reader.
Page 8, line 248-255: Due to the similarity of the “Young people” and “Young adult” categories, and knowing that there is also a reference of a “middle aged” group previously, it gets confusing to understand this sentence. I highly suggest clarifying the age groups.
Page 9, line 303: Your study has not only limitations but also strengths. It would be interesting to have a paragraph on the strengths of your study!
Author Response
We would like to thank the editor and reviewers for their comments in this review, which have greatly improved the readability of the manuscript. We would like to inform you that we have edited the manuscript according to the very constructive suggestions from the reviewers.
Below, please find a list of revisions and a response to each of the reviewer’s comments. We hope that the revisions in the manuscript and our accompanying responses will be enough to make our manuscript suitable for publication in the IJERPH.
We shall look forward to hearing from you at your earliest convenience.
Yours sincerely,
The authors

Reviewer 2 Report
Authors: Congratulations to the authors for the work done in this research and for addressing such a current and relevant topic.
The following are some comments for the authors' assessment:
ABSTRACT
- Line 21. The sentence "An increase in the difficulty of performance and a decrease in satisfaction with it." seems loose. It should be linked to the previous sentence.
INTRODUCTION
- Line 54. There is much more literature to support this statement: "Previous research [18] support a relationship between OB, health and life satisfaction."
MATERIALS ANS METHODS
- 2.2. Participants. The procedure needs to be better explained. If data were collected in a single time period, it would be necessary to explain how the period before the pandemic was evaluated and to justify why it was done in this way. In addition, the discussion should explain why this was done when some questionnaires (such as the OBQ) are not intended to be used in this way (it asks questions in relation to the last week).
Signing the consent form is not a criterion for inclusion in this study per se. It is mandatory for participation in any study.
In exclusion criterion (2) there seems to be a typo ("por"). Also, this criterion could be explained a little more.
Results" and part of its content (lines 86-88) have been introduced under "2.2. Participants".
- Data Collection. In relation to the OBQ, why has the Spanish version of the OBQ (OBQ-E) not been used when in the introduction several Spanish studies have been referenced in which this version has been used (references 15 and 18)? (references 15 and 18). In those studies the OBQ-E is used and properly referenced: Peral Gómez, P. Occupational Balance in Occupational Therapy Students. Ph.D. Thesis, Universidad Miguel Hernández, Alicante, Spain, 2017. Also available reference: Peral-Gómez P, López-Roig S, Pastor-Mira MÁ, Abad-Navarro E, Valera-Gran D, Håkansson C, Wagman P. Cultural Adaptation and Psychometric Properties of the Spanish Version of the Occupational Balance Questionnaire: An Instrument for Occupation-Based Research. Int J Environ Res Public Health. 2021 Jul 14;18(14):7506. doi: 10.3390/ijerph18147506. In addition, when discussing any of the instruments, mention should be made of their good psychometric properties.
Regarding the "Questions on the modification of routine and habits", what Likert-type scale has been followed, are all questions closed-ended, yes/no or are there more ranges and/or scores? It is not clear.
- 2.4. Data Analysis. Categorical variables should be expressed in absolute frequencies and percentages. When continuous variables do not meet normality, they should be expressed with median and interquartile range. Why has the Kolmogorov-Smirnov test not been used to assess whether the normality criteria are met? The statistical significance value that was established for the analysis is missing. What analyses have been done to compare the measurements at the 2 time points?
RESULTS
- The final sample consisted of 300 participants. Has a previous sample calculation been made to know what would be a sufficient n for this study, taking into account that the main variable is OB? The question arises as to whether 300 participants are representative.
- In Table 1, the third age group (65 years and older) has very few subjects compared to the rest. There would have to be more n in this group or divide the groups in a more balanced way to make better analyses. The same is true for the "north" and "south" groups of the region of residence. Comparing a group of 19 subjects with a group of 213 is not recommended.
- Regarding the paragraph "When analyzing the differences between the variables before the pandemic and post-confinement period of pandemic, significant differences were found in all variables, with values of greater impact in those referring to post-confinement period of pandemic. In other words, greater occupational imbalance (z= -10.789; p < 0.01) was found." (lines 147-150) It gives the impression that these analyses are not explained in the "Data Anylisis" section. Also, in all comparisons was the same value found? It is not clear because it is not explained.
- It would be interesting to also include (in Table 2 or in the text) the total OBQ score of the total sample before the pandemic and during the pandemic and to know if there are differences.
- Regarding the paragraph "Regarding the qualitative variables on habits and routines, significant differences were found in most of the variables, with a greater number of the sample having modified their routines (p<0.01), lost habits (p<0.01) and not resumed habits (p<0.01)" (lines 165-167). It is not clear what has been analyzed here (between what are the significant differences? between groups?). This should be explained and could be accompanied by a table with the results.
- Regarding the sentence "When analyzed by age group, young people (18-30 years), young adults (31-64 years) 174 and older adults (65 years and older), differences in habits were found (Figure 2)." (lines 174-175) Were these differences significant or not? It is not clear because the data is not provided.
- Regarding the sentence "Regarding performance and satisfaction, when analyzing the differences between the variables before the pandemic and current period, significant differences were found in all variables, with" (lines 179-180) It would be advisable to be able to see these data.
- Figure 3 Would it be possible to better define the colors in the figure? They are very difficult to differentiate.
DISCUSSION
- Regarding the sentence "and the existence of differences between people who have had COVID-19 and those who have not." (lines 212-213) This has not been analyzed or mentioned in the results.
- In relation to the sentence "Some of the activities were reduced, as well as the time spent 215 on them." (lines 215-216) This has been measured in any way? I don't know if it can be deduced from the data collected. If so, it should be explained in the methodology and results.
- Regarding the sentence "Savitsky et al. (2021) [25] found a positive correlation between occupational satisfaction, the importance of the activities in which they participated, workload, professional support and psychological reward." (lines 223-225) In relation to what results is this reference provided?
- The corresponding reference is missing from the sentence "Nevertheless, these data are at odds with pre-pandemic research, which reported no significant sex differences." (lines 228-229).
- Lines 229-232. The information on the differences between positive or negative COVID-19 is interesting but has not been studied in this research.
Author Response
We would like to thank the editor and reviewers for their comments in this review, which have greatly improved the readability of the manuscript. We would like to inform you that we have edited the manuscript according to the very constructive suggestions from the reviewers.
We want to attach an excel sheet with some of the requested data, but it only allows us to upload one file (the response letter). We will try to send it to the editor. We hope the editor will send it to you.
Please find a list of revisions and a response to each of the reviewer’s comments. We hope that the revisions in the manuscript and our accompanying responses will be enough to make our manuscript suitable for publication in the IJERPH.
We shall look forward to hearing from you at your earliest convenience.
Yours sincerely,
The authors

Reviewer 3 Report
Dear authors
Thank you for submitting this manuscript. The subject matter is timely and interesting, however, significant improvements are needed, especially when it comes to grammer and spelling. Please see my comments below that may be of some help.
Abstract
Line 14, put "the" before "COVID-19 pandemic"
Line 16, remove "... after the pandemic beginning ..." and add instead "... post the beginning of the pandemic ..."
In line 17, add "the" before "Occupational Balance Questionnaire ..."
You mention the Occupational Balance Questionnaire in the abstract, however, I have noted the use of multiple questionnaires in the methods. You need to mention these questionnaires in the abstract also.
Line 25, put "... was ..." after "There" and before "decrease"
In the keywords, make sure there is a capital B at "balance;"
Introduction
On line 31, you have reference [1,2] straight after "The World Health Organisation (WHO)..." It makes more sense to have the reference at the end of the sentence in this instance.
Line 34 - change sentence so it says "... alterations may significantly affect ..." instead of "... alterations significantly may affect ..."
In line 37 - before "Covid passport" put "the requirement of COVID..." additionally, Covid should be COVID and this needs to be consistent throughout the text.
Also in line 37, remove "to" after "access" and remove "or" before "hospitals"
Line 38 - remove "etc" as this is not acceptable in academic writing. Instead use "and so on." Additionally, the next word should just say "Now" and not "Nowadays,"
Line 39 - remove "social distancing" and replace with "restrictive measures" Additionally, remove "requirements (indoor - outdoor)" and replace with "wearing"
In line 40 - remove "from the viral spread." and replace with "cross contamination"
Line 41 - remove "The h..." and replace with "Home ..." Additionally, put a comma after "confinement" and change "have" to "has"
Line 42 - after [8-10] add "which" before "negatively" Remove "ing" from "impacting" and add an "s" so it reads "impacts"
Line 51 - and elsewhere, you mention he or she. It is now accepted that there are other forms of gender identity so please replace any mention of he or she to they. This is to be done unless there is clear evidence that no one defined themselves as they or any other gender comformity now accepted.
Line 54 - you mention previous research, but only give one reference, there needs to be more references here to document the extent of previous research.
Line 57, and else where, you mention et al. followed by the year. This is not congruient with the author guidelines for this journal and as such any reference to a piece of work in text that has the year beside it most be removed.
Line 58 - replace not after However, with no
In line 60/61 delete a after over and before longer, add an s after period and delete of time
Line 63 - delete 's beginning and work on sentence so it states "... after the beginning of the pandemic. Additionally, delete the so it reads "... compared to that of pre-pandemic periods..."
After period place a full stop and then start the next sentence with: "Additionally, this study will also determine the existence ..."
2.1 Design
Firstly, a paragraph should be made up of more than one sentence
Secondly, the paragraph starting on line 68 and ending in line 70 needs to be expanded more. Some topics to discuss here is what is a cross-sectional, descriptive design? what are the STROBE checklist and why was it deemed suitable for this study?
The second paragraph in design talks of research ethics, may I suggest for the revised manuscript that this is a sub-heading of design and from which you expand on the ethics a tad bit more
2.2 Participants
Line 79 - remove "... were ..." and replace with "as"
On line 80/81 you give a time period of approximately 3 weeks and not a month as suggested - this is misleading and needs to be rectified.
One again line 79-81 is only one sentence. This needs to be expanded more. Tell the reader what is snowball non-probability sampling. Additionally, ensure you review this and see as I am nearly certain it can only be used in qualitative research and not quantitative research.
You mention as part of the inclusion/exclusion criteria that you are only including men and women, what about the non-binary community - why are they not represented? You need to be very clear here. Additionally, I feel there is not enough in the inclusion/exclusion criteria to effectively narrow the pool. But maybe it is your intention to have it wide enough for everyone to participate - if this is the case, I am even more puzzled as to why the non-binary community were not consulted.
In line 85 - please remove the bolded 3. Results
Line 86 - 88 should not be included in the text as this is guidance for how to fill in the section. This is also at your data availability statement and at the beginning of the references section of this text.
From line 94 - 126, you discuss some of the tools used to measure aspects of OB. These need to be under the sub-heading measures and then have their own minor headings where there is much more discussion as to how these tools were created, by whom, what they are used for, what strenghts/limitations are associated with such tools and are they well validated - with some referenced examples.
From line 116 - 126 you suggest that you have made your own scale - I am unsure as to whether you can add your own one along with two validated scales. Additionally, the fact that this was made by you and from what I can see not validated suggests to me that the inclusion of this weakens the study and therefore needs to be acknowledged in the strenghts/limitations section of the discussion
Line 117 - the word "stile" should be "style" delete is beside question and replace with posed was ...
Line 118 - delete the global pandemic of ...
Line 124 - 126, align with remainder of the paragraph
2.4 Data Analysis
Please expand on this section considerably so you can explain the tests carried out during data analysis.
3. Results
Line 141 - please expand what you mean by "missing data"
Line 148 - add "the" between "of" and "pandemic"
Line 154 - Delete "is" between "analysis" and "performed" and replace with "was"
Line 165 - you mention "qualitative" for the first time here. I assume this is an error. Otherwise, the fact that it is a variable makes the study more mixed method than quantitative and if so the manuscript in its entirety will need to be revised.
For all figures, please supply a colour copy of same as it is difficult to ascertain the different variables here.
For all figures - the "figure title needs to be above the figure and not below same
Line 180 - please revise as this study is not in the current period as it was conducted in February/March 2021.
4. Discussion
Line 209 - remove "main" after "the" and before "purpose"
Line 211 - add "... beginning of the pandemic ..." and remove the "beginning" after "pandemic" Additionally remove "also" and replace with "Additionally this study was conducted ..."
Line 213 - you mention "differences between people who have had COVID-19 and those who have not." This was not investigated in the study so therefore needs to be removed as it is misleading.
Line 214 - remove "year of"
Line 219 - remove "the study" and replace with "... that found in a study ... "
Line 220 - remove "... in which ..." and add "where"
In line 228-229 - the sentence starting with "Nevertheless ..." and ending in "... sex differences" contradicts the sentence pre-ceeding it.
Line 235 - delete "... according ..." and replace with "based on"
Line 237 - put "the" between "of" and "pandemic"
Line 238 - you mention "In contrast to our results, other authors [26] ..." You only give one reference here, however, the way the sentence is presented, we need to see more references that can justify the statement made.
Line 239 - change "these" to "the", change "are" to "is"
Line 242 - the sentenced should read : ... articles supporting the fact that these variables ..."
Line 244 - delete "global"
Line 255 - Delete "global" and "caused by COVID-19" so that it reads "... a result of the COVID-19 pandemic ..."
Line 257: add "as" after "pandemic"
Line 259 - delete "... regions prior to the pandemic ..." and replace with "... two time periods ..."
Line 264-266 should be under the heading recommendations for future research
In line 269, remove "the" beside "analyzed" and "job"
Line 273, remove "... in their study".
Line 280 - remove "of the?" beside "Most" and "participants"
Line 282 - delete "these" and "are" so that the sentence reads: "... such data is consistent ..."
In line 285 - delete"it was" and replace with "they"
Line 288/89 - add "related" between "COVID-19" and "confinement"
Line 291 - 300 should be under the heading limitations of the study
Line 301 - 303 should be under recommendations for future study
5. Conclusion
Please re-examine the conclusion and add more information to it.
Overall, the study has merit but needs serious restructuring as well as grammer and spelling rechecks to be at a standard suitable for publication. I look forward to reading your revised manuscript.
Author Response
We would like to thank the editor and reviewers for their comments in this review, which have greatly improved the readability of the manuscript. We would like to inform you that we have edited the manuscript according to the very constructive suggestions from the reviewers.
Please find a list of revisions and a response to each of the reviewer’s comments. We hope that the revisions in the manuscript and our accompanying responses will be enough to make our manuscript suitable for publication in the IJERPH.
We shall look forward to hearing from you at your earliest convenience.
Yours sincerely,
The authors

Round 2
Reviewer 1 Report
Thank you so much to the authors for considering my comments.
I believe the manuscript is ready to be published in its current form.
Best regards
Author Response
Thank you so much for your support.
Reviewer 2 Report
First of all, I would like to thank the authors for the work done to incorporate the comments of revision 1. It is a very interesting research and the clarifications add quality to the manuscript.
With the intention of improving some more details, I would like to make some comments for the assessment of the authors:
ABSTRACT
- Lines 18-20. It seems that in these lines "besides questions about satisfaction and performance of an activities and questions on the modification of routine and habits. Also, satisfaction and occupational performance, as well as the modification of routines and habits were administered" say the same thing, but in different ways. It is advisable to avoid these repetitions.
MATERIALS ANS METHODS
- 2.3.1. Outcome Measures. In relation to the OBQ, on line 99, the reference of the original questionnaire should appear. And when OBQ-E is indicated, on line 104-105, the reference corresponding to that version (this already appears correctly).
In relation to the "Questions on the modification of routine and habits", it is not clear which Likert-type scale has been followed nor what the questions are like (are all the questions closed, yes/no or are there more ranges and/or scores?).
- 2.4. Data Analysis. It is not clear whether the continuous variables meet or do not meet normality. If they do not meet normality they should be expressed with median and interquartile range.
RESULTS
- Table 2 should show Median (IQR) or M (SD), depending on whether the variable is normal or not. But both data should not appear.
- Line 181. There is a typo "foundbetween". The space is missing.
- In Figure 1 there is a 188. What does it mean, is it a typo?
- In relation to the results of performance and satisfaction, in addition to the figures, the data of the significant differences found should appear.
CONCLUSIONS
- The conclusions state that "This study contributes to improve the intervention of people with immunodeficient pathologies, which due to their clinical conditions also require processes of social isolation". However, this has not been the objective of this research and has not been analyzed, so nothing should be concluded in this regard.
If it is considered interesting, the idea could be introduced in the discussion section, perhaps as a future line of research and justifying the idea.
Author Response
We would like to thank the reviewer for giving us the opportunity for a second round.
We hope that the revisions in the manuscript and our accompanying responses will be enough to make our manuscript suitable for publication in the IJERPH.
We shall look forward to hearing from you at your earliest convenience.
Yours sincerely,
The authors
